

# Identifying Chern numbers of superconductors from local measurements

Paul Baireuther[1], Marcin Płodzień[2], Teemu Ojanen[3,4],
Jakub Tworzydło[5] and Timo Hyart[2,6,7]

**1** Bosch Center for Artificial Intelligence,
Robert-Bosch-Campus 1, 71272 Renningen, Germany
**2** International Research Centre MagTop, Institute of Physics, Polish Academy of Sciences,
Aleja Lotników 32/46, PL-02668 Warsaw, Poland
**3** Computational Physics Laboratory, Physics Unit, Faculty of Engineering and
Natural Sciences, Tampere University, PO Box 692, FI-33014 Tampere, Finland
**4** Helsinki Institute of Physics, PO Box 64, FI-00014 Helsinki, Finland
**5** Faculty of Physics, University of Warsaw, ulica Pasteura 5, 02-093 Warszawa, Poland
**6** Department of Applied Physics, Aalto University, 00076 Aalto, Espoo, Finland
**7** Computational Physics Laboratory, Physics Unit, Faculty of Engineering and
Natural Sciences, Tampere University, PO Box 692, FI-33014 Tampere, Finland

## Abstract

**Fascination in topological materials originates from their remarkable response properties and exotic quasiparticles which can be utilized in quantum technologies. In particular, large-scale efforts are currently focused on realizing topological superconductors and their Majorana excitations. However, determining the topological nature of superconductors with current experimental probes is an outstanding challenge. This shortcoming has become increasingly pressing due to rapidly developing designer platforms which are theorized to display very rich topology and are better accessed by local probes rather than transport experiments. We introduce a robust machine learning protocol for classifying the topological states of two-dimensional (2D) chiral superconductors and insulators from local density of states (LDOS) data. Since the LDOS can be measured with standard experimental techniques, our protocol contributes to overcoming the almost three decades standing problem of identifying the topological phase of 2D superconductors with broken time-reversal symmetry.**



# 1   Introduction

The problem of determining the topological states of superconductors dates back three decades to the discovery of $Sr_2RuO_4$. Despite the abundance of accumulated data, the topological state of $Sr_2RuO_4$ remains under lively debate [1–7]. Time-reversal breaking 2D topological insulators and superconductors are classified by an integer-valued Chern number $C$ [8–10] and thus the determination of the topological state boils down to the identification of it. In contrast to insulators, where $C$ determines the value of quantized Hall conductance [8], in superconductors it gives rise to quantized thermal Hall conductance [11, 12]. This difference is at the heart of the problem of determining $C$ of a superconductor. While the quantization of electronic Hall conductance has been observed in remarkable accuracy [13], thermal conductance measurements in superconductors have not reached the required sophistication to observe the quantization.

While it is unclear whether 2D topological superconductivity exists in any naturally occurring material, there is accumulating evidence that it can be realized in the lab [14–16]. The various topological designer platforms can realize rich topological phase diagrams [17–19], and this is exemplified in a Shiba lattice formed by magnetic impurities on a superconducting surface (Fig. 1(a)), which supports a large number of topologically distinct phases (Fig. 1(b)) [17]. A recent experimental study of a Shiba lattice found signatures consistent with chiral Majorana edge modes and the accompanying theory calculation suggested that the system could be in a $C \sim 20$ state [15]. However, the value of the $C$ could not be experimentally confirmed, because of the absence of a diagnostic tool to experimentally access the Chern number.

The long-standing impasse in the determination of topological invariants in superconductors has also become an obstacle in developing quantum technologies. Topological superconductors can harbor unpaired Majorana states at vortex cores, point defects and edge vortices, and they could be employed in quantum computing [12, 20–22]. However, the lack of a robust and unbiased protocol for the identification of the topological state is hindering the progress of the field and has led to many debates about the interpretation of the experiments.

For the above reasons, it is imperative to devise a generic protocol for extracting Chern numbers of superconductors from experimentally feasible data. Recently, it has been demonstrated, that machine learning can be used to discriminate quantum phases of matter, based on theoretically simulated data [23–25] or combinations of theoretical and experimental data [26]. Unsupervised methods [27] and combinations of supervised and unsupervised methods [28] are other interesting directions in the classification of phases of quantum matter and determination of phase transitions, for an extended review see [29]. However, to our knowledge, so far machine learning methods have not been applied to determine the Chern number from data which is accessible with standard experimental techniques in superconductors.

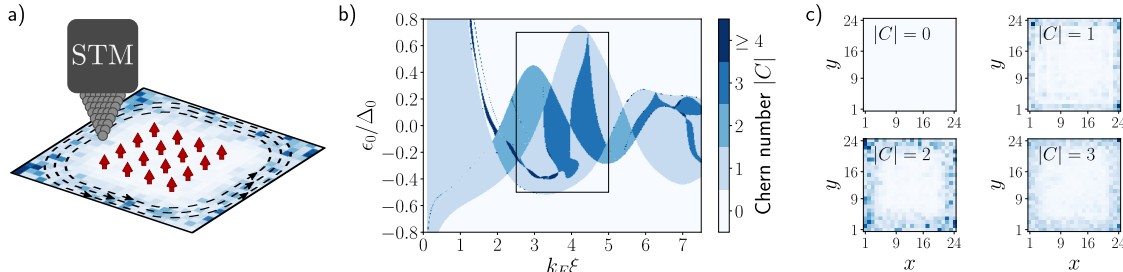

Figure 1: (a) Illustration of a magnetic impurity lattice immersed in a $s$-wave superconductor which supports topological phases with high Chern numbers $C$. The modulus of the Chern number $|C|$ determines the number of chiral edge modes showing up in the LDOS measured using an STM tip. (b) Phase diagram of the Shiba lattice model as a function of Fermi momentum $k_F$ and Shiba energy $\epsilon_0$ expressed in units of coherence length $\xi$ and superconducting bulk gap $\Delta_0$, respectively. We have assumed a square lattice of impurities with a lattice constant $a = \xi$ and a Rashba spin-orbit coupling $\lambda = \alpha_R/(\hbar v_F) = 0.1$, where $v_F$ is the Fermi velocity. The box indicates the region of parameters used for creating the dataset for testing the neural network predictions. (c) Random examples of the LDOS for various $|C|$ in the presence of strong disorder $V_0 = 0.8\overline{\Delta}_{\text{shiba}}$, where $\overline{\Delta}_{\text{shiba}}$ is the average bulk gap in the Shiba model dataset. The LDOS is averaged over a symmetric energy window around zero energy $[-\overline{\Delta}_{\text{shiba}}/6, \overline{\Delta}_{\text{shiba}}/6]$, and therefore, the tunneling density of states is proportional to the quasiparticle density of states. Hence, STM measurements in the weak coupling limit will produce the type of pictures illustrated here.

In this work we present a protocol which is using local density of states (LDOS) data as input and supervised machine learning to extract the topological state information. In our numerical experiments, we train ensembles of artificial neural networks with generic tight-binding Hamiltonians and show that the topological phases with Chern number moduli $0, 1, 2, 3$ of a Shiba lattice system are classified correctly with a high probability ($> 96\%$) in a representative part of the phase diagram. Since the LDOS data is accessible with standard scanning tunneling microscopy (STM), our work constitutes an important step towards solving the long-standing problem of identifying the topology of 2D superconductors with broken time-reversal symmetry. Moreover, we describe how the reliability of the predictions can be estimated, and we find, that when the criteria for a reliable prediction are satisfied, the prediction accuracy reaches values exceeding 99%.

## 2 Shiba lattice model and test dataset

To demonstrate the performance of our machine learning assisted Chern number extraction protocol, we identify the Chern numbers of a complex Shiba lattice system from its LDOS data. The Shiba model describes a superconducting surface decorated with a two-dimensional lattice of ferromagnetic adatoms as illustrated in Fig. 1(a). This model contains the generic ingredients of topological superconductivity and supports a complex pattern of topological phases as a function of the system parameters [17]. For the applicability of the machine learning protocol, however, the detailed assumptions of the model and its relationship to a specific physical system are not important. The Bogoliubov-de Gennes (BdG) Hamiltonian for the bulk electrons in the Nambu basis $(\hat{\Psi}_\uparrow, \hat{\Psi}_\downarrow, \hat{\Psi}_\downarrow^\dagger, -\hat{\Psi}_\uparrow^\dagger)$ is

$$\mathcal{H}_{\mathbf{k}}^{(\text{bulk})} = \tau_z\big[\xi_{\mathbf{k}}\sigma_0 + \alpha_R(k_y\sigma_x - k_x\sigma_y)\big] + \Delta_0\tau_x\sigma_0\,, \tag{1}$$

where $\xi_{\mathbf{k}} = \frac{\hbar^2 k^2}{2m} - \mu$ is the kinetic energy, $\mu$ is the chemical potential, $\alpha_R$ is the Rashba spin-orbit coupling strength, $\Delta_0$ is the superconducting pairing amplitude, and $\tau_i$ and $\sigma_i$ are the Pauli matrices for the particle-hole and spin degrees of freedom, respectively. The electrons couple to the magnetic impurities by the exchange interaction $J$ as

$$\mathcal{H}^{(\mathrm{imp})}(\mathbf{r}) = -J \sum_j \mathbf{S}_j \cdot \boldsymbol{\sigma} \, \delta(\mathbf{r} - \mathbf{r}_j), \tag{2}$$

where $\mathbf{r}_j$ are the positions of the magnetic moments and $\mathbf{S}_j$ their spins. The total Hamiltonian is given by $\mathcal{H} = \mathcal{H}^{(\mathrm{bulk})} + \mathcal{H}^{(\mathrm{imp})}$. A single magnetic impurity binds one fermionic state at energy $\epsilon_0$ within the superconducting gap, and in the derivation of the effective Hamiltonian we consider the limit $\epsilon_0 \ll \Delta_0$ for simplicity. The coupling between the impurity states at $\mathbf{r}_i$ and $\mathbf{r}_j$ is described by effective hopping $h_{ij}$ and pairing $\Delta_{ij}$ amplitudes, which decay as $1/\sqrt{r_{ij}}$ at short distances and exponentially at distances larger than the superconducting coherence length $\xi$ of the substrate (see Sec. 6.1). By transforming the Hamiltonian to momentum space we obtain

$$
\begin{aligned}
H(\mathbf{k}) &= \mathbf{d}(\mathbf{k}) \cdot \boldsymbol{\tau}, \\
d_x(\mathbf{k}) &= \mathrm{Re} \sum_{ij} e^{-i\mathbf{k}\cdot\mathbf{r}_{ij}} \Delta_{ij}, \\
d_y(\mathbf{k}) &= -\mathrm{Im} \sum_{ij} e^{-i\mathbf{k}\cdot\mathbf{r}_{ij}} \Delta_{ij}, \\
d_z(\mathbf{k}) &= \sum_{ij} e^{-i\mathbf{k}\cdot\mathbf{r}_{ij}} h_{ij},
\end{aligned}
\tag{3}
$$

where $\mathbf{r}_{ij} = \mathbf{r}_i - \mathbf{r}_j$. The effective Hamiltonian $H(\mathbf{k})$ generally defines a gapped band structure and satisfies particle-hole symmetry $\tau_x H(\mathbf{k})^* \tau_x = -H(-\mathbf{k})$. Thus the model belongs to the Altland-Zirnbauer class $D$ and the topological phases are classified by Chern numbers

$$C = \frac{1}{4\pi} \int_{\mathrm{BZ}} d^2 k \, \hat{\mathbf{d}} \cdot \left( \frac{\partial \hat{\mathbf{d}}}{\partial k_x} \times \frac{\partial \hat{\mathbf{d}}}{\partial k_y} \right), \quad \hat{\mathbf{d}} = \mathbf{d}/|\mathbf{d}|. \tag{4}$$

In all calculations we assume a square lattice of impurities with a lattice constant equal to the superconducting coherence length $a = \xi$. While in typical experiments the lattice constant is shorter than the coherence length, we choose this slightly longer lattice constant to obtain a simple phase diagram containing large patches of topological phases. Moreover, in experiments the magnetic atoms are typically placed on the surface of a 3D superconductor, reducing the coupling between the Shiba states in comparison to our 2D model calculations. Therefore, the magnitudes of the effective couplings between the Shiba states in our model are probably quite realistic although we set the distance between the impurity atoms larger than in experiments. We set a Rashba coupling $\lambda = \alpha_R/(\hbar v_F) = 0.1$, where $v_F$ is the Fermi velocity in the absence of spin-orbit coupling. The qualitative features of the phase diagram do not depend on the exact value of $\lambda$ [30]. Under these constraints the low-energy theory is fully defined by two dimensionless parameters $\epsilon_0/\Delta_0$ and $k_F \xi$, where $k_F$ is the Fermi wave vector in the absence of Rashba coupling (see Sec. 6.1). By varying these two parameters we obtain a rich topological phase diagram, and in the following we concentrate on a representative portion of the parameter space containing topological phases with $|C| = 0, 1, 2, 3$, as shown in Fig. 1(b). The modulus of the Chern number $|C|$ determines the number of chiral edge modes, and therefore the LDOS data contains information about $|C|$ but it reveals nothing about the sign of $C$. We calculate LDOS maps for systems of size $N_x = N_y = 24$ lattice sites averaged over an energy window $[-\overline{\Delta}_{\mathrm{shiba}}/6, \overline{\Delta}_{\mathrm{shiba}}/6]$, where $\overline{\Delta}_{\mathrm{shiba}}$ is the average bulk gap in the Shiba dataset.

We calculate $\overline{\Delta}_{\text{shiba}}$ by first determining the average gap of the clean and infinite systems separately for each Chern number $|C|$ and then taking the mean. In these calculations we have also introduced a disorder potential sampled from uniform distributions $[-V_0, V_0]$. To avoid excessive finite size effects, we generate the data by considering only systems where the bulk gap is at least $3.5/N_x \approx 0.15$ and the strength of the disorder potential satisfies $V_0 \leq \overline{\Delta}_{\text{shiba}}$ (see Sec. 6.3). As shown in Fig. 1(c), the nontrivial phases $|C| \neq 0$ can be easily distinguished from the trivial phase $C = 0$ by the LDOS at the boundaries due to the edge modes. However, whether one can identify the value of $|C|$ only from the LDOS, which can be accessed by standard experimental methods, is a highly nontrivial question. Below we demonstrate that a high-accuracy determination of $|C|$ is possible using a machine learning-assisted protocol.

# 3 Machine learning-assisted determination of the Chern number

Topological properties of physical systems are global in nature and cannot be deterministically identified by a local measurement. However, it remains an open problem how reliably a topological state can be inferred from local data, such as the LDOS, measured across *the whole sample*. This data contains information about (i) the non-local correlations of quasiparticle states at the opposite sides of the sample that are utilized in various topological tests and (ii) interference patterns that are known to be different for the unconventional pairing states where the Cooper pairs carry finite angular momenta. The intimate relationship of the Chern number, topological edge states, and Cooper pair angular momenta suggests that the LDOS contains information about the Chern number.

In this work we employ artificial neural networks for the identification of the Chern number from LDOS data while making only rough assumptions about the underlying system. For this purpose a LDOS map is propagated through the neural networks, which output a "prediction vector" assigning a weight ("predicted probability") to each element in a predefined set of Chern number moduli (see Fig. 2). We use convolutional neural networks (CNNs) [31], because they are efficient in identifying features in the interference patterns independently of their spatial positions in the LDOS maps. Our neural network architecture is described in Sec. 6.4, but we emphasize that our results do not depend strongly on the details of the network as long as it is expressive enough.

To train the neural networks we employ well established supervised learning techniques. A loss function compares the neural network's predictions to the labels of the training data, and we utilize stochastic gradient descent in the parameter space of the neural network with the objective of minimizing the loss. While this approach works well across a large range of applications [32], it relies on the existence of the training data. We generate a labeled training dataset, consisting of LDOS maps and corresponding Chern numbers, by using a distribution over all low-energy models which fulfill the necessary symmetry requirements and other relevant physics motivated constraints. This is tractable, because the low-energy physics of topological superconductors and insulators can typically be described reasonably well with simple models, where only the quasiparticle bands closest to the Fermi energy are taken into account. We discuss the details of these models in Sec. 6.2.

In typical machine learning problems, the training data is sampled from the same distribution as the test data to which the neural network will be applied after the training. Therefore, when the *average* error on the training data decreases smoothly, the same is true for the average error on the test data if the data sets are large enough so that overfitting is not a problem. In our case, however, the test data is generated using a distribution $\mathcal{D}_t$ of Shiba lattice model Hamiltonians whose support is a tiny fraction of the support of the training model Hamiltonians' distribution $\mathcal{D}$. As a consequence, even if the prediction error (i.e. the generalization

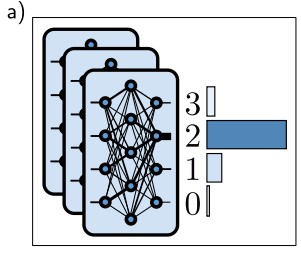
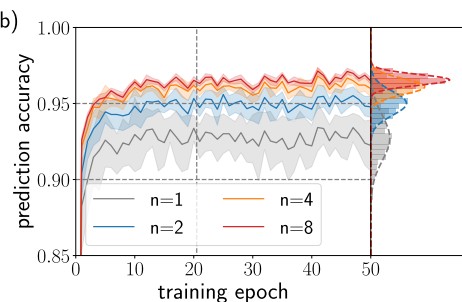
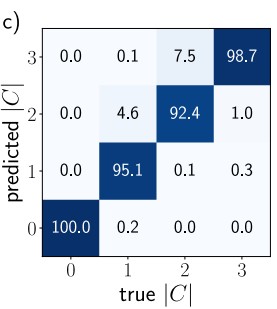

Figure 2: (a) Schematic depiction of an ensemble of neural networks. Each ensemble member gets as input a LDOS map and outputs a prediction vector assigning a weight to each element in a set of predetermined possible values of the Chern number modulus. These prediction vectors are then averaged, resulting in a joint prediction vector which is schematically illustrated by the histogram. (b) Demonstration of the advantage of ensembles over individual neural networks. Gray: Performance of a single neural network. In color: Performance of ensembles of size $n > 1$. The mean prediction accuracy (line) and the standard deviation (shaded region) have each been calculated for sets of 13 unique ensembles drawn randomly out of a pool of 13 individually trained networks. The histograms on the right summarize the statistics of the prediction accuracy over the training epochs 21-50. The average accuracies of the ensembles over these epochs are $n = 1 : 0.927 \pm 0.015$, $n = 2 : 0.950 \pm 0.008$, $n = 4 : 0.961 \pm 0.005$, and $n = 8 : 0.966 \pm 0.004$. (c) Confusion matrix for ensemble size $n = 8$ averaged over 13 unique ensembles and the training epochs 21-50.

error) averaged over the training distribution decreases smoothly, the prediction error on the test set can, and in our numerical experiments does, fluctuate wildly between adjacent training epochs (see Fig. 9). Minimizing these fluctuations is paramount for robust identification of the Chern number and for being able to decide when to stop the training of the neural networks without access to the test data.

To solve this problem we employ not just a single neural network, but an ensemble of neural networks [33, 34]. Each network is trained individually on a slightly different training data set, and then the prediction vectors are averaged to give a collective prediction vector. The averaging effect of the ensemble reduces the fluctuations between training epochs strongly.

Moreover, if two Chern numbers are equally possible for a given LDOS we expect that the average prediction vector gives roughly equal weights to both Chern numbers, and we can exploit this knowledge by discarding all predictions where the ensemble is divided between multiple Chern numbers. This reduces the number of miss-classified examples and allows us to estimate the reliability of each individual prediction.

## 4 Results

The results of our numerical experiments are summarized in Figs. 2 and 3. To evaluate the robustness of the performance, we trained the networks for many more epochs than necessary and evaluated them on the Shiba dataset after each epoch. We find that there is a dramatic advantage of using an ensemble of networks instead of a single network (see Fig. 2 b)). The ensemble does not only average out the fluctuations (see Fig. 9) in the prediction accuracy but it also has a higher prediction accuracy than any of the individual networks on its own. This counter-intuitive result occurs because during ensemble averaging the uncorrelated part of the error in the predictions of the individual networks averages out to a certain degree [33, 34].

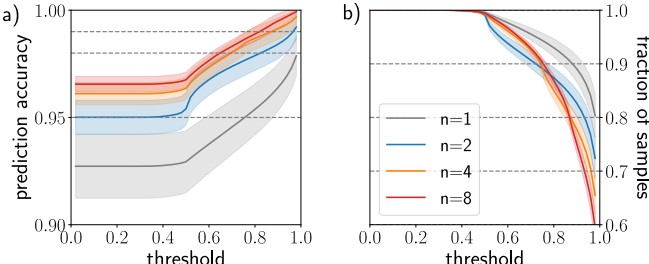

Figure 3: (a) Prediction accuracy as a function of certainty threshold. (b) Fraction of data for which the certainty threshold is met. In both panels, we show curves for the same ensemble configurations as in Fig. 2. The mean and standard deviation are obtained as in the histograms of Fig. 2.

In addition, we empirically see that networks predicting the Chern number correctly tend to be more confident about their predictions (the prediction vector has a very large weight on the correct Chern number) than the networks which are predicting the Chern number incorrectly (the prediction vector typically has significant weight on at least two Chern numbers). In the ensembles this effect is pronounced. As can be seen in Fig. 2 the ensemble of networks predicts the Chern number correctly with a very high average probability > 96%. Furthermore, in the case of a sufficiently large ensemble the prediction accuracy varies very little during the epochs 21-50. Therefore, the utilization of the ensemble of networks also solves the problem of deciding when to stop training the networks without access to the test data, because the training can be stopped at any time after the networks have been trained sufficiently. The general convergence and stopping point of the neural network training can be determined using only the numerically generated data and standard methods from machine learning, such as early stopping [35]. The confusion matrix shows that the ensemble of networks predicts all Chern numbers $|C| = 0, 1, 2, 3$ correctly with high accuracy > 92% (see Fig. 2(c)).

The prediction vector also allows us to gain insights into the reliability of the predictions. Namely, we can consider the largest weight in the prediction vector divided by the sum of all weights as a measure of the certainty of the prediction. Fig. 3 shows that the prediction accuracy indeed increases significantly if we introduce a certainty threshold larger than 0.5 and discard the samples that do not satisfy this criterion. In the case of reliable predictions, where the ensembles are almost certain about their predictions, the average accuracy of the identification reaches values exceeding 99%.

The use of an ensemble of neural networks, and the observation that the networks predicting the Chern number correctly from the LDOS tend to be more confident about their predictions, makes our protocol applicable to realistic systems. It is of course advantageous to generate the training data from a distribution that has large weight on the models expected to describe the physical system, and therefore knowledge of the system can be utilized in the sampling of the training Hamiltonians, but it is also important to sample them from a sufficiently broad distribution to make sure that the system of interest is covered by the distribution over training Hamiltonians. Therefore, the uncertainties related to the physical system determine the optimal broadness of the distribution of the training models. Our numerical experiments resemble a realistic setting because we have introduced only rough global constraints which could be deduced from experimental systems. In our numerical experiments, we employed a very broad distribution of training models compared to the distribution of Shiba models to demonstrate that a detailed knowledge about the physical system is not needed. In the case of our numerical experiments, the prediction accuracy converges as a function of the size of the ensemble already at approximately $n \sim 10$ networks, but we expect that larger ensembles are needed in the case of broader distributions of the training models. In this work, the collective

prediction vector was obtained by averaging the prediction vectors of the ensemble but for a broader distribution of the training models it may be beneficial to give larger weight on the confident predictions.

We emphasize that our protocol has fundamental limitations in the sense that it cannot determine the Chern numbers of all mathematically possible class D Hamiltonians, which is also not our goal because almost all of these Hamiltonians strongly violate the symmetries, smoothness or locality constraints of physical systems. Rather, it provides a practical approach for classification of topological states, in particular in topological designer platforms. Because the key property of topological superconductors is an integer-valued topological invariant which is insensitive to many system details, one might expect that the amount of experimentally accessible information can be sufficient for probabilistic determination of the invariant, and our protocol provides a way to efficiently extract and utilize the essential information for that purpose.

In real systems the shapes and sizes of the samples will not be exactly as planned, and it can therefore be beneficial to generate training data using various shapes and sizes of samples and then use padding to make the input fit the neural networks' requirements, or to introduce strong disorder near the system boundary. Moreover, away from the boundary the low-energy LDOS of a topological superconductor is fairly featureless, and therefore one could probably transform the measured LDOS map of a physical system in a smart way to fit the input size of the neural network by cutting and reconstructing an LDOS map of suitable size.

## 5 Discussion

In this work, we have outlined a protocol for extracting the Chern number of superconductors via machine learning assisted analysis of the LDOS data. In our numerical experiments, we showed that an ensemble of neural networks, trained using generic tight-binding Hamiltonians, can identify the topological phases with $|C| = 0, 1, 2, 3$ in a representative part of the Shiba lattice model phase diagram, away from the phase boundaries, correctly with an average probability of $> 96\%$. Since the LDOS can be measured with standard experimental techniques, our protocol constitutes an important step in overcoming the long-standing problem of identifying the topology of 2D superconductors. In the future it would be important to extend testing of the protocol beyond the single model and restricted parameter space used in this work.

The key advantage of our method is that, in the generation of the training data, we only need very rough information on the Hamiltonian corresponding to the system of interest. This allows us to parametrize a distribution of training models which covers the low-energy physics of this system as a special case. Importantly, as illustrated in our numerical experiments, the set of possible models describing the system can be a tiny fraction of the distribution of the training models. For these reasons our method works well even if there exist large uncertainties regarding the details of the studied system and sets it apart from earlier work [25] which required specific knowledge about the physical system and was hence not applicable to real physical systems.

The ultimate goal of our research is to make reliable predictions of the Chern number from experimental data. We expect our method could be applied almost directly to many of the experimentally realized designer platforms because they are expected to share many qualitative features of the Shiba lattice models. For future research it would be interesting to generalize our method to higher Chern numbers and to explore its limits. Another interesting avenue of research would be to utilize additional experimental data, in particular non-local transport measurements, where two scanning tunneling microscope tips are used instead of

one. We expect that this could help to improve the accuracy and robustness of the predictions and it would allow the determination of the sign of the Chern number. Finally, we expect that our method could be utilized also in the determination of other topological invariants and employed in quality control of topological materials and quantum devices.

# 6 Methods

## 6.1 Details of the Shiba lattice model

To make our analysis self-contained, we summarize here the details of the Shiba lattice model derived in Ref. [17]. Starting from the BdG Hamiltonian $\mathcal{H} = \mathcal{H}^{(\text{bulk})} + \mathcal{H}^{(\text{imp})}$, described in the main text, the equation for the eigenstate $\Psi$ at energy $E$ can be written as

$$\left[E - \mathcal{H}^{(\text{bulk})}(\mathbf{r})\right]\Psi(\mathbf{r}) = -J\sum_j \mathbf{S}_j \cdot \boldsymbol{\sigma}\, \delta(\mathbf{r} - \mathbf{r}_j)\Psi(\mathbf{r}). \tag{5}$$

This equation can be solved for $\Psi(\mathbf{r})$ using Fourier transform yielding

$$\Psi(\mathbf{r}) = -\sum_j J_E(\mathbf{r} - \mathbf{r}_j)\hat{S}_j \cdot \boldsymbol{\sigma}\, \Psi(\mathbf{r}_j), \tag{6}$$

with $S = |\mathbf{S}|$, $\hat{S} = \mathbf{S}/S$ and the integral

$$J_E(\mathbf{r}) = JS \int \frac{d\mathbf{k}}{(2\pi)^2} e^{i\mathbf{k}\cdot\mathbf{r}}\left[E - \mathcal{H}_{\mathbf{k}}^{(\text{bulk})}\right]^{-1}.$$

The propagator can be written as the sum of two helical components $\left[E - \mathcal{H}_{\mathbf{k}}^{(\text{bulk})}\right]^{-1} = \frac{1}{2}(G_- + G_+)$, where

$$G_{\pm} = \frac{\left(E\tau_0 + \xi_{\pm}\tau_z + \Delta_0\tau_x\right)\left(\sigma_0 \pm \sin\varphi\,\sigma_x \mp \cos\varphi\,\sigma_y\right)}{E^2 - \xi_{\pm}^2 - \Delta_0^2},$$

with $\xi_{\pm} = \xi_k \pm \alpha_R k$ and $\mathbf{k} = k(\cos\varphi, \sin\varphi)$.

For the case of a single impurity that is deep inside the gap $|E| < \Delta_0$ it is possible to determine the integration over $J_E(\mathbf{0})$ which leads to an equation for the wave-function at $\mathbf{r} = \mathbf{0}$

$$\left[\mathbb{1} - \frac{\alpha}{\sqrt{\Delta_0^2 - E^2}}(E\tau_0 + \Delta_0\tau_x)\hat{S}\cdot\boldsymbol{\sigma}\right]\Psi(\mathbf{0}) = 0,$$

where $\alpha = \pi JS\mathcal{N}$ is a dimensionless quantity characterizing the impurity strength depending on the density of states at the Fermi level $\mathcal{N} = \frac{1}{2\pi}\frac{m}{\hbar^2}$ in the absence of Rashba coupling. From this equation, the states induced by the single impurity can be determined. They are given by $|\tau_x+\rangle|\uparrow\rangle$ with eigenvalue $E = \Delta_0\frac{1-\alpha^2}{1+\alpha^2}$ and $|\tau_x-\rangle|\downarrow\rangle$ with eigenvalue $E = -\Delta_0\frac{1-\alpha^2}{1+\alpha^2}$. The actions of the spin operator are $\hat{S}\cdot\boldsymbol{\sigma}|\uparrow\rangle = |\uparrow\rangle$ and $\hat{S}\cdot\boldsymbol{\sigma}|\downarrow\rangle = -|\downarrow\rangle$, and the operator $\tau_x$ acts in the Nambu space as $\tau_x|\tau_x\pm\rangle = \pm|\tau_x\pm\rangle$. In the following, we denote the energy of an isolated impurity state as $\epsilon_0 = \Delta_0\frac{1-\alpha^2}{1+\alpha^2}$ and concentrate on the limit $\alpha \approx 1$ so that $\epsilon_0 \approx \Delta_0(1-\alpha)$.

For a lattice of impurities where the impurity separation $a$ is large enough so that $E \ll \Delta_0$ for all impurity band energies, the impurity band can be found from equation

$$\left[\mathbb{1} - \left(\frac{E}{\Delta_0}\tau_0 + \alpha\,\tau_x\right)\hat{S}_i\cdot\boldsymbol{\sigma}\right]\Psi(\mathbf{r}_i) = -\sum_{j\neq i}\lim_{\substack{E\to 0 \\ \alpha\to 1}} J_E(\mathbf{r}_i - \mathbf{r}_j)\hat{S}_j\cdot\boldsymbol{\sigma}\,\Psi(\mathbf{r}_j). \tag{7}$$

In the special case that the impurity lattice is a ferromagnetic arrangement of spins $\mathbf{S}_i = S\hat{e}_z$, Eq. (7) can be projected on the decoupled Shiba states $|\tau_x+\rangle|\uparrow\rangle$ and $|\tau_x-\rangle|\downarrow\rangle$. It then follows that

$$J_{E=0}(\mathbf{r}) = -\frac{\alpha}{2}\Big\{\big[I_1^-(\mathbf{r}) + I_1^+(\mathbf{r})\big]\Delta_0\tau_x\sigma_0 - \big[I_2^-(\mathbf{r}) - I_2^+(\mathbf{r})\big]\tau_z\sigma_x$$
$$+ \big[I_3^-(\mathbf{r}) - I_3^+(\mathbf{r})\big]\tau_z\sigma_y\Big\} + g(\tau_z, \sigma_{x/y}, \tau_x\sigma_{x/y}),$$

$$I_1^\pm(\mathbf{r}) = \frac{1}{2\pi^2}\frac{\mathcal{N}_\pm}{\mathcal{N}}\int d\varphi \int d\xi \, \frac{e^{ik^\pm(\xi)r\cos\beta}}{\Delta_0^2 + \xi^2},$$

$$I_2^\pm(\mathbf{r}) = \frac{1}{2\pi^2}\frac{\mathcal{N}_\pm}{\mathcal{N}}\int d\varphi \int d\xi \, \frac{e^{ik^\pm(\xi)r\cos\beta}\,\xi\sin\varphi}{\Delta_0^2 + \xi^2},$$

$$I_3^\pm(\mathbf{r}) = \frac{1}{2\pi^2}\frac{\mathcal{N}_\pm}{\mathcal{N}}\int d\varphi \int d\xi \, \frac{e^{ik^\pm(\xi)r\cos\beta}\,\xi\cos\varphi}{\Delta_0^2 + \xi^2},$$

where $\mathcal{N}_\pm = \mathcal{N}\big[1 \mp \lambda/\sqrt{1+\lambda^2}\big]$, $\lambda = \alpha_R/(\hbar v_F)$, $v_F$ is the Fermi velocity in the absence of Rashba coupling, $k^\pm(\xi) = k_F^\pm + \xi/(\hbar\tilde{v}_F)$, $k_F^\pm = k_F\big(\sqrt{1+\lambda^2} \mp \lambda\big)$, $\tilde{v}_F = v_F\sqrt{1+\lambda^2}$, $r = |\mathbf{r}|$, $\beta$ is the angle between $\mathbf{r}$ and $\mathbf{k}$, and $g(\tau_z, \sigma_{x/y}, \tau_x\sigma_{x/y})$ denotes terms proportional to $\tau_z\sigma_0, \tau_0\sigma_x, \tau_0\sigma_y, \tau_x\sigma_x$, or $\tau_x\sigma_y$ whose matrix elements between the low-energy basis states vanish. The solution of the integrals can be expressed in terms of Bessel $J_n$ and Struve functions $H_n$, yielding

$$I_1^\pm(\mathbf{r}) = \frac{\mathcal{N}_\pm}{\mathcal{N}}\frac{1}{\Delta_0}\text{Re}\Big[J_0\big(k_F^\pm r + ir/\xi\big) + iH_0\big(k_F^\pm r + ir/\xi\big)\Big],$$

$$I_2^\pm(\mathbf{r}) \equiv i\,I_0^\pm(r)\sin\varphi', \quad I_3^\pm(\mathbf{r}) \equiv i\,I_0^\pm(r)\cos\varphi',$$

$$I_0^\pm(\mathbf{r}) = \frac{\mathcal{N}_\pm}{\mathcal{N}}\text{Re}\Big[iJ_1\big(k_F^\pm r + ir/\xi\big) + H_{-1}\big(k_F^\pm r + ir/\xi\big)\Big],$$

where $\mathbf{r} = r(\cos\varphi', \sin\varphi')$ and $\xi = \hbar\tilde{v}_F/\Delta_0$. Assuming that the impurities are far away from each other, we can use the asymptotic expressions

$$I_0^\pm(\mathbf{r}) \approx -\frac{\mathcal{N}_\pm}{\mathcal{N}}\Bigg[\sqrt{\frac{2/\pi}{k_F^\pm r}}\sin\big(k_F^\pm r - \tfrac{3\pi}{4}\big)e^{-r/\xi} + \frac{2/\pi}{(k_F^\pm r)^2}\Bigg],$$

$$I_1^\pm(\mathbf{r}) \approx \frac{\mathcal{N}_\pm}{\mathcal{N}}\frac{1}{\Delta_0}\sqrt{\frac{2/\pi}{k_F^\pm r}}\cos\big(k_F^\pm r - \tfrac{\pi}{4}\big)e^{-r/\xi}. \tag{8}$$

Projecting on the low-energy states yields an effective Hamiltonian

$$\sum_j \begin{pmatrix} h_{ij} & \Delta_{ij} \\ \Delta_{ji}^* & -h_{ij} \end{pmatrix}\begin{pmatrix} u(\mathbf{r}_j) \\ v(\mathbf{r}_j) \end{pmatrix} = E\begin{pmatrix} u(\mathbf{r}_i) \\ v(\mathbf{r}_i) \end{pmatrix}, \tag{9}$$

where the two components of the wave functions are $u(\mathbf{r}_j) \equiv \langle\tau_x+|\langle\uparrow|\Psi(\mathbf{r}_j)\rangle$ and $v(\mathbf{r}_j) \equiv \langle\tau_x-|\langle\downarrow|\Psi(\mathbf{r}_j)\rangle$, and the entries of the Hamiltonian are given by

$$h_{ij} = \begin{cases} \epsilon_0, & i = j, \\ -\dfrac{\Delta_0^2}{2}\big[I_1^-(r_{ij}) + I_1^+(r_{ij})\big], & i \neq j, \end{cases}$$

$$\Delta_{ij} = \begin{cases} 0, & i = j, \\ \dfrac{\Delta_0}{2}\big[I_0^+(r_{ij}) - I_0^-(r_{ij})\big]\dfrac{x_{ij} - iy_{ij}}{r_{ij}}, & i \neq j. \end{cases}$$

In all numerical calculations we use the asymptotic expressions for $I_{0,1}^\pm(\mathbf{r})$.

## 6.2 Details of the training model Hamiltonians

The low-energy physics of topological superconductors and insulators can always be described with models, where only the quasiparticle bands closest to the Fermi energy are taken into account. The Hamiltonian $H(\mathbf{k})$, describing the lowest bands of a particle-hole symmetric $\tau_x H^*(\mathbf{k})\tau_x = -H(-\mathbf{k})$ system, has a generic form

$$H(\mathbf{k}) = \begin{pmatrix} \xi(\mathbf{k}) & \Delta(\mathbf{k}) \\ \Delta^*(\mathbf{k}) & -\xi(-\mathbf{k}) \end{pmatrix}, \quad \Delta(-\mathbf{k}) = -\Delta(\mathbf{k}). \tag{10}$$

Because it is beneficial to have large energy gaps, we consider models obeying inversion and mirror symmetries (w.r.t. the high-symmetry axes and the diagonal)

$$\xi(k_x, k_y) = \sum_{\sqrt{n_x^2 + n_y^2} \leq d} \xi_{n_x, n_y} e^{i(n_x k_x + n_y k_y)},$$

$$\xi^*_{n_x, n_y} = \xi_{n_x, n_y}, \ \xi_{-n_x, n_y} = \xi_{n_x, n_y},$$

$$\xi_{n_x, -n_y} = \xi_{n_x, n_y}, \ \xi_{n_y, n_x} = \xi_{n_x, n_y},$$

and

$$\Delta(k_x, k_y) = \sum_{\sqrt{n_x^2 + n_y^2} \leq d} \Delta_{n_x, n_y} e^{i(n_x k_x + n_y k_y)},$$

$$\Delta_{-n_x, n_y} = \Delta^*_{n_x, n_y}, \ \Delta_{n_x, -n_y} = -\Delta^*_{n_x, n_y},$$

$$\Delta_{n_y, n_x} = -i\Delta^*_{n_x, n_y}.$$

We choose the cut-off in the hopping distance as $d = 2$ and sample the Hamiltonians from a distribution where the hopping amplitudes decrease with the distance as

$$\xi_{n_x, n_y} = \tilde{\xi}_{n_x, n_y} e^{-\frac{\sqrt{n_x^2 + n_y^2}}{\xi}}, \qquad \Delta_{n_x, n_y} = \tilde{\Delta}_{n_x, n_y} e^{-\frac{\sqrt{n_x^2 + n_y^2}}{\xi}},$$

and $\tilde{\xi}_{n_x, n_y} \in [-1, 1]$, $|\tilde{\Delta}_{n_x, n_y}| \in [0, 1]$ and $\arg(\tilde{\Delta}_{n_x, n_y}) \in [0, 2\pi]$ are random variables restricted by the symmetry requirements. For the decay length $\xi$ in the training models it is beneficial to use an estimate based on the studied system, so that in our case we set $\xi = 1$ corresponding to the coherence length $\xi = a$ in the Shiba lattice model. We calculate the LDOS maps for systems of size $N_x = N_y = 24$ lattice sites averaged over an energy window $[-\overline{\Delta}_{\text{training}}/6, \overline{\Delta}_{\text{training}}/6]$, where $\overline{\Delta}_{\text{training}}$ is the average bulk gap in the training dataset. We generate the training data by considering only systems where the bulk gap is at least $3.5/N_x \approx 0.15$ and the strength of the disorder potential satisfies $V_0 \leq \overline{\Delta}_{\text{training}}$ (see Sec. 6.3). To be computationally efficient and since high precision is not important, we determine the bulk gap size only with an average accuracy of $\approx 99\%$. Because of the requirement of large energy gaps, there is a tendency of generating examples of strong coupling superconductivity. To avoid this problem, we only accept samples with superconducting order parameters describing weak or moderate coupling

$$\sqrt{\sum |\Delta_{n_x, n_y}|^2} \leq \sqrt{\sum |\xi_{n_x, n_y}|^2 - |\xi_{0,0}|^2}.$$

## 6.3 Data generation protocol

We generate clean system Hamiltonians from the distributions described in Secs. 2 and 6.2, calculate their Chern numbers $C$ and energy gaps, and pre-select subsets so that the bulk gap is at least $3.5/N_x$. For each subset we calculate the Chern number dependent average bulk gap

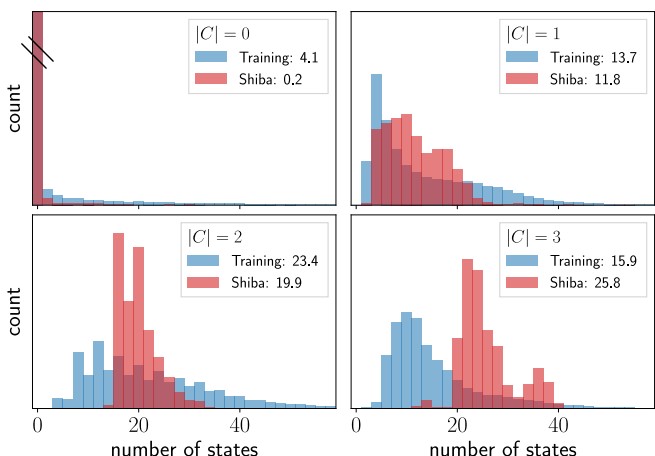

Figure 4: Histograms showing the number of states in the energy window $[-\bar{\Delta}_\nu/6, \bar{\Delta}_\nu/6]$ ($\nu$ = training, shiba) for $|C| = 0, 1, 2, 3$ on an arbitrary scale. The average number of states for each $|C|$ is shown in the legend of the corresponding figure. For most samples the number of states is significantly larger than $2|C|$ which is the minimal requirement to be able to detect $|C|$ from the LDOS data.

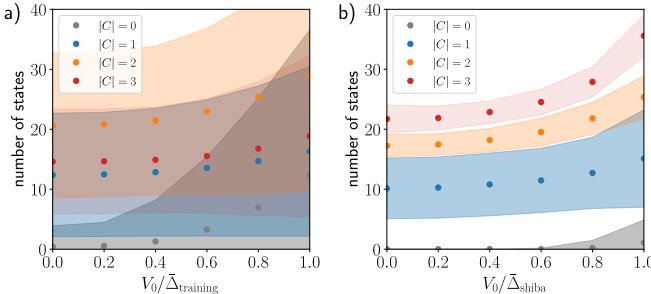

Figure 5: Average number of states in the energy window $[-\bar{\Delta}_\nu/6, \bar{\Delta}_\nu/6]$ ($\nu$ = training, shiba) for $|C| = 0, 1, 2, 3$ as a function of disorder amplitude. The dots denote average values and the colored regions depict one standard deviation around the averages. Left: Training models; Right: Shiba models.

$\bar{\Delta}_{\nu,|C|}$ ($\nu$ = training, shiba) and define $\bar{\Delta}_\nu$ as their respective average. Note that according to our convention the gap appears in the energy interval $E = [-\Delta/2, \Delta/2]$.

We discretize these Hamiltonians on a lattice of size $N_x = N_y = 24$ and add an onsite disorder potential sampled from uniform distributions $[-V_0, V_0]$ with $V_0/\bar{\Delta}_\nu = \{0, 0.2, 0.4, 0.6, 0.8, 1.0\}$. We verified the construction of the training tight-binding Hamiltonians using an alternative implementation based on the KWANT toolbox [36]. In the case of the Shiba models, we obtain a balanced dataset by choosing the number of disorder realizations per Hamiltonian such that the resulting dataset has approximately the same number of samples for each Chern number. The Shiba dataset contains 11k datapoints in total [37]. In the case of the training data we choose the same number of Hamiltonians for each Chern number and then use one disorder configuration per Hamiltonian and disorder strength resulting in 75k data points for each Chern number $|C| = 0, 1, 2, 3$. In addition we generate a smaller validation dataset with 7.5k data points per Chern number. For each disorder realization, we calculate the LDOS

$$\rho(x, y) \propto \sum_{E_i \in \mathcal{E}} |\langle x, y | \psi_i \rangle|^2, \tag{11}$$

using an energy window $\mathcal{E} = [-\bar{\Delta}_\nu/6, \bar{\Delta}_\nu/6]$ which is chosen so that we have a reasonable

number of edge states without getting too many disorder-induced bulk states (see Figs. 4 and 5). We normalize the LDOS maps so that the standard deviation across each sample is one, except if there are no states in the LDOS window, then all values of the LDOS map are zero. This is not only beneficial for the neural network training, but also removes the ambiguity of the typically unknown prefactor in LDOS measurements.

We can study the effects of the finite size of the samples and the disorder on the topology by computing the Chern marker $C_m$ integrated over the bulk of the sample [38–40]. If the system is sufficiently large to possess the topological properties $C_m$ is approximately quantized to the integer value of the Chern number of an infinite system. Fig. 6(a) shows the modulus of the Chern marker $|C_m|$ as a function of the system size for a representative set of system parameters. Our choice of the system size $N_x = N_y = 24$ yields a relatively good quantization in all considered cases. Also, manufacturing systems of size $24\xi$ does not pose a significant experimental challenge. Fig. 6(b) shows the Chern marker $C_m$ as a function of disorder amplitude $V_0$ for the same set of system parameters demonstrating that $|C_m|$ is reasonably well quantized in the range of disorder amplitudes considered in our work $V_0/\overline{\Delta}_{\text{shiba}} \leq 1$. We also roughly estimated the average localization lengths of the edge states ($\sim 2$ lattice constants in the Shiba dataset and $\sim 4$ lattice constants in the training dataset) to check that they are much smaller than the system size.

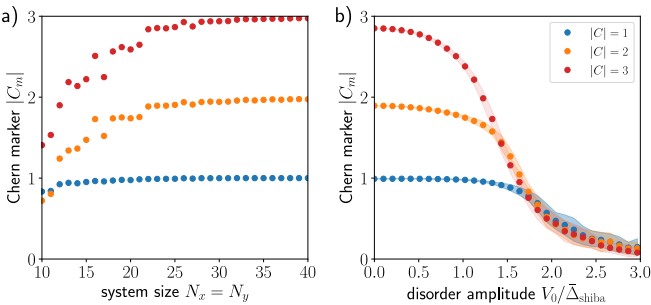

Figure 6: (a) Chern marker of clean Shiba lattice model systems as a function of system size. (b) Chern marker for a system size $N_x = N_y = 24$ as a function of disorder amplitude $V_0/\overline{\Delta}_{\text{shiba}}$ (averaged over 20 disorder realizations). The parameters are $k_F \xi = 3.68$, $\epsilon_0/\Delta_0 = 0.35$ for $C = 1$, $k_F \xi = 2.88$, $\epsilon_0/\Delta_0 = 0.29$ for $C = 2$, and $k_F \xi = 3.49$, $\epsilon_0/\Delta_0 = 0.16$ for $C = 3$.

## 6.4 Details of the neural network and robustness to the neural network architecture

Our neural networks take as input an LDOS image $\rho$ and output an estimated probability distribution $p(|C|\,\big|\rho, \theta)$ over the Chern numbers $|C| \in \{0, 1, 2, 3\}$, conditional on $\rho$ and the trainable parameters (weights) $\theta$. We have used several different neural network architectures to test the robustness of our results (see Table 1). To build, train and evaluate the networks, we used the KERAS library [41] with TENSORFLOW [42] as backend and the ADAM optimization algorithm [43]. The source code we used to generate the training data and to train and evaluate the neural network ensembles can be found at [44]. For each network an individual training dataset was created by sampling with redraw a dataset of the same size from the 300k training data points. The respective 300k training data points were randomly shuffled before each epoch and fed into the optimizer in mini-batches of 64 samples. The validation dataset was used to detect training convergence and overfitting (see Fig. 9). In the following, we explain the operation of the different layers appearing in our neural networks.

Table 1: The different neural network architectures considered for determination of the Chern number. The naming of the layers is mostly based on the conventions used in KERAS and their operation are discussed in the text. The results reported in the main text are obtained using networks with the "CNN 64" architecture, but we have checked that all networks give similar results and show the results from the other network architectures in Figs. 7 and 8. The last row of the table shows the average accuracies for ensembles of size $n = 8$.

| name | CNN 32 | CNN 64 | CNN double layer 64 |
|---|---|---|---|
| layers | RandomFlip | RandomFlip | RandomFlip |
| | Conv2D (32x3x3) | Conv2D (64x3x3) | Conv2D (64x3x3) |
| | | | Conv2D (64x3x3) |
| | MaxPooling2D (32x2x2) | MaxPooling2D (64x2x2) | MaxPooling2D (64x2x2) |
| | Conv2D (32x3x3) | Conv2D (64x3x3) | Conv2D (64x3x3) |
| | | | Conv2D (64x3x3) |
| | MaxPooling2D (32x2x2) | MaxPooling2D (64x2x2) | MaxPooling2D (64x2x2) |
| | Flatten | Flatten | Flatten |
| | Dense (32) | Dense (64) | Dense (64) |
| | Dense (4) | Dense (4) | Dense (4) |
| weights | 47k | 185k | 259k |
| avg. accuracy | $96.1 \pm 0.3$ | $96.6 \pm 0.4$ | $96.3 \pm 0.4$ |

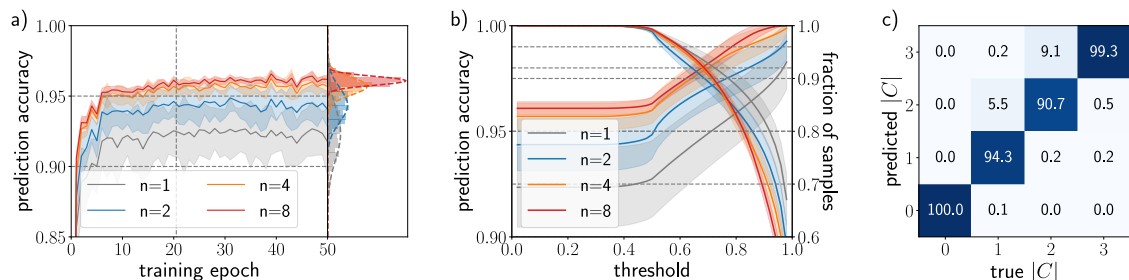

Figure 7: Same as Figs. 2 and 3 but for the network "CNN 32" as detailed in Table 1.

CNNs [31] are built on the idea that patterns can be efficiently identified independently on their spatial position in the image. This is done using "filters" which traverse over the entire image considering only a small window of pixels at a time to estimate if a certain feature is present. The output of a convolutional layer is again an image comprising $N$ values "channels" for each pixel, one for each filter. We use the notation, Conv2D ($N \times w \times w$), where $N$ is the number of channels and $w \times w$ is the kernel size, i.e. the size of the input window of the filters.

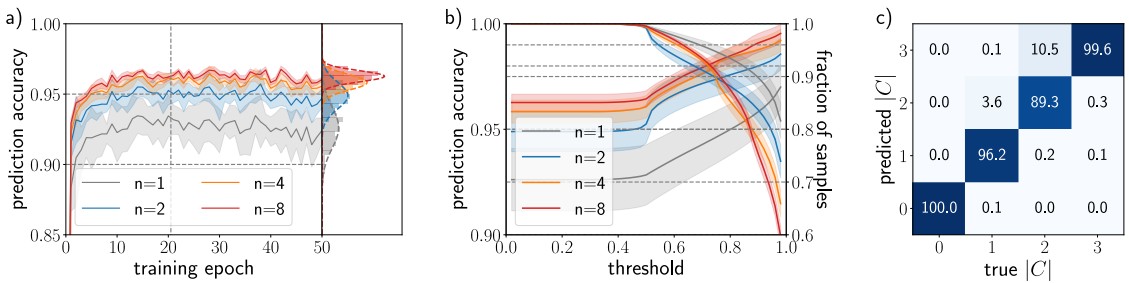

Figure 8: Same as Figs. 2 and 3 but for the network "CNN double layer 64" as detailed in Table 1.

We use stride length 1 (step size in the movement of the filter), and padding (inclusion of rows and columns of zeros around the image) such that the size of the output image is the same as the size of the input image. After each convolutional layer, we apply an element-wise "rectified linear" activation function

$$\phi(x) = \begin{cases} 0, & x < 0, \\ x, & x \geq 0, \end{cases}$$

to the output image.

The "pooling" layers reduce the dimensionality of the image by transforming the information in a small window of pixels into a single pixel. This operation acts independently on each channel, and we use so called "max pooling" which returns the maximal value in each pooling window. We use the notation MaxPooling2D ($N \times w \times w$), where $N$ is the number of channels and $w \times w$ is the kernel size. We use stride length 2 and no padding, so that the image size is reduced by a factor of 2 in each direction.

Additionally, we also have a fully connected layer denoted as Dense ($N$), where $N$ is the number of rectified linear units (i.e. neurons) in the layer.

The other layers are used for technical purposes. The RandomFlip layers augment the data by randomly flipping the image w.r.t the $x$- and the $y$-axis and is only used during training and the Flatten layers turn a tensor into a vector.

In the last layer, denoted as Dense (4), we apply a linear mapping from the feature vector $f \in \mathbb{R}^N$ to a 4-vector

$$q\big(|C| \big| \rho, \theta\big) = \sum_{j=1}^{N} A_{|C|j} f_j(\rho) + b_{|C|},$$

with one entry for each allowed $|C| = 0, 1, 2, 3$, with trainable parameters $A_{|C|j} \in \mathbb{R}^{4 \times N}$ and $b_{|C|} \in \mathbb{R}^4$.

Finally, we use the softmax function

$$[\text{softmax}(x)]_i = \frac{\exp(x_i)}{\sum_{j=1}^{K} \exp(x_j)}, \quad \text{for } x \in \mathbb{R}^K,$$

to squeeze $q$ into a vector resembling a probability distribution

$$p\big(|C| \big| \rho, \theta\big) = \text{softmax}\big(q\big(|C| \big| \rho, \theta\big)\big),$$

with $p\big(|C| \big| \rho, \theta\big) \geq 0$ and $\sum_{|C|} p\big(|C| \big| \rho, \theta\big) = 1$. As a loss function we use the categorical cross-entropy given by

$$\mathcal{L} = - \sum_{(\rho, |C|) \in \mathcal{D}_{\text{mini-batch}}} \log(p(|C| \big| \rho; \theta)).$$

## 6.5 Details of the neural network ensembles

The ensembles in the main text consist of two or more neural networks which are trained individually using different initial weights and biases. In addition, each network is trained using a slightly different dataset. These datasets are sampled from the original dataset with redraw and are of the same size as the original dataset. This way of constructing an ensemble is called bagging [34].

As a computationally cheaper variant, one can also use "snap shots" of a single neural network at different training times. In Fig. 10 we show this for the case where $n$ snapshots of the same neural network at adjacent training epochs form an ensemble of strongly correlated

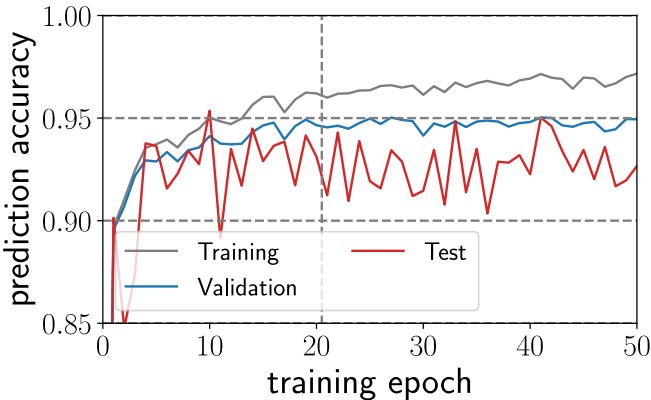

Figure 9: Exemplary training curve of one of the networks used in the main text. In gray, accuracy of the network evaluated on the training data. In blue, accuracy of the network evaluated on unseen "validation" data from the same distribution as the training data. Such validation data is accessible during training and can be used to decide when to stop training. The performance gap with respect to the training data indicates some overfitting. We therefore expect that with more training data our results could be improved further. In red, accuracy of the network evaluated on the Shiba test data, which can not be used for performing or terminating the training process. The large fluctuations between adjacent training epochs illustrate the need for reducing the variance of the generalization using ensembles.

networks. As for the ensembles in the main text, the fluctuations between adjacent training epochs are strongly reduced. However, the average performance is lower than that of the ensembles of independently trained neural networks and the variance across the ensembles is greater. In addition, we observe that the performance declines slightly after hitting its peak during the training progresses. This is likely because as the training progresses, the changes in the neural network at adjacent training epochs become smaller and smaller and therefore the networks in the ensemble become more and more correlated.

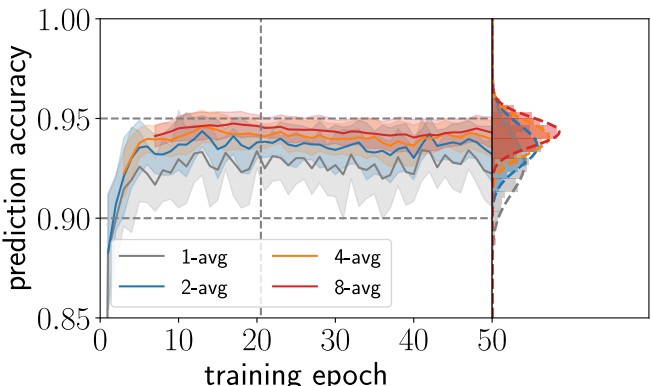

Figure 10: Same as Fig. 2, but instead of $n$ independent networks, here each ensemble consists of $n$ instances of the same network at adjacent training epochs ("running average"). The mean prediction accuracy (line) and the standard deviation (shaded region) have been calculated using the same 13 neural networks as in Fig. 2. Average accuracy of $n = 1 : 0.927 \pm 0.015$, $n = 2 : 0.936 \pm 0.011$, $n = 4 : 0.940 \pm 0.008$, and $n = 8 : 0.943 \pm 0.007$.

# Acknowledgments

**Funding information** M.P. and T.H. were supported by the Foundation for Polish Science through the IRA Programme co-financed by EU within SG OP. T.O. acknowledges project funding from the Academy of Finland and Helsinki Institute of Physics.

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
