# Peer review of "Identifying Chern numbers of superconductors from local measurements"

_SciPost Physics, doi:SciPost Phys. Core 6, 087 (2023)_

## Round 1 · Referee Report · Adrian Del Maestro (Referee 1) · 2022-6-21

Strengths

  1. Introduction is well written, with a concise yet useful discussion of the main focus of the paper -- the utility of the Chern number in characterizing quantized thermal conductance in topological superconductors.
  2. 95% success rate of supervised learning approach in identifying topological phases of the Shiba lattice.
  3. Ensemble ML approach utilizing multiple networks is used, with the outputs averaged giving a collective predictive vector identifying the probability of a Chern number that is robust against strong fluctuations during training due to rare signals in the training data.
  4. The machine learning protocol, especially the use of an ensemble approach is well described, with careful analysis of the statistical accuracy of ensemble predictions.
  5. The manuscript includes some interesting ideas for addressing issues (non-uniformity, sparse data, etc.) that would naturally appear when training on real experimental LDOS data.
  6. Figures are clear and informative.
  7. Detailed model architecture included in Table I.

Weaknesses

  1. Not clear why the specific value of $\lambda = 0.1$ is chosen in the model.
  2. Assumption of the model is that the lattice spacing between impurity states is exactly the superconducting coherence length $\xi$.
  3. Shiba Lattice Model section lacks sufficient details for the reader. While many of these are buried in a methods section at the end of the paper, important physical quantities like the $V_0$, the strength of the disorder potential appear out of nowhere.
  4. Is $24\times \xi$ a reasonable field of view in terms of a real experiment?
  5. Claim in the discussion: "our protocol essentially overcomes the long-standing problem of identifying the topology of 2D superconductors." is likely overselling. Only a single model has been considered, in some restricted region of parameter space on mostly idealized data. The approach is interesting and promising, and will likely spur further work.
  6. Data set and code are not included or made available with the manuscript. This will hinder reproducibility and potentially impact of the work.

Report

The authors report on a supervised machine learning approach to classify topological states of 2D chiral superconductors and insulators using local density of states (LDOS) data. The method is tested on the Shiba lattice, an engineered quantum system formed by arranging magnetic impurities on a superconducting surface. Synthetic data is generated using an effective Hamiltonian that couples a simple BdG superconductor to magnetic impurities. By using an ensemble of networks to make predictions, good accuracy (over 90%) is achieved in classifying Chern numbers up to |C| = 3.

Overall the individual sections of the manuscript are clearly written with descriptive figures. The inclusion of detailed methods, only as a sort of appendix at the end reduces readability and make it more difficult to get a picture of the model and dataset before the results are presented. The choice of model parameters, and their connection to real engineered quantum systems is also lacking. However, the results are promising and will hopefully spur further work using real experimental data. As such, I believe this work opens a new pathway in an existing research direction, the identification of unambiguous signatures of topological superconductors. The lack of data or code with the manuscript is a drawback in this sense, and will make follow ups by the community more difficult.

Requested changes

  1. Page 1: "... imperative to device ..." → "... imperative to devise"
  2. Enhance the level of detail in the "Shiba Lattice Model and Test Data Set" section. For example, while a picture of a STM spectra is shown in Figure 1a), more discussion is needed on why this is the expectation from an experiment. The sentence: "The modulus of the Chern number |C| determines the number of chiral edge modes, and therefore the LDOS data contains information about |C| ..." should be expanded upon do better explain how STM spectra is sensitive to edge modes.
  3. Appearance of disorder potential scale $V_0$ needs to be better explained earlier in the manuscript. The reader sees it 2 times before it is eventually explained on page 7.
  4. Tone down claim in discussion regarding solution to the problem of identifying topology of 2D superconductors.
  5. Extra space should be removed in spin kets in top paragraph in the RHS column of page 6.
  6. What is "respectively" referring to in the following sentence "at the bulk gap is at least $3.5/N_x \approx 0.15$ and 0.15, respectively" on page 7?
  7. Page 8, "CNNs [23] are build ..." → "CNNs [23] are built ..."

---

## Round 2 · Referee Report · Anonymous (Referee 2) · 2023-11-8

Report

The manuscript studies whether a supervised ML algorithm can be useful in identifying the Chern number of a Shiba lattice using LDOS. The manuscript correctly states that LDOS in principle does not contain sufficient information to identify the Chern number, and therefore argues that using a combination of knowing the conceptual model describing the Shiba lattice together with a trained ML model is up to the task. This is supported by examining the generalizability of the model outside of the training dataset. This validation, however, is insufficient for the algorithm to be applicable to a real world scenario: a real life system may have more dramatic biases, such as different spin order, terms omitted from an effective Hamiltonian, spatial inhomogeneity, and a significantly larger disorder. Furthermore, the manuscript does not address the question how one would determine whether in a specific experiment one could defend that the proposed approach has a sufficient generalizability. For that reason I believe that the manuscript details a useful advance in ML identification of topological phases. It, however, does not address the problem it is set to resolve—namely the identification of the topological phases based on the real world data. Therefore I consider the manuscript suitable for SciPost Physics Core rather than SciPost Physics.

---

## Round 2 · Author Response

Resubmission Letter

We thank the editor for considering our manuscript and for providing additional references to put our work in a larger context.

We thank the referee for his valuable feedback based on which we have revised the manuscript.

Response to referee report

Dear Prof. Del Maestro,

Thank you very much for your valuable feedback, based on which we have revised our manuscript. We have addressed most of your comments and suggestions directly in the manuscript and have summarized those changes in the "List of changes". In addition, in the manuscript we have provided references to the source code we used to train and evaluate the neural network ensembles as well as to the Shiba lattice model dataset. In the following, we would like to address the remaining points.

  1. Regarding our choice of model parameters, and their connection to real engineered quantum systems:

    1.1 In [1], it was shown that the Shiba lattice model phase diagrams are qualitatively similar for a range of values of \(\lambda\). We only require that \(\lambda\) is not too small, because it controls the size of the bulk energy gap. Otherwise, the exact value of \(\lambda\) is not important.

    1.2 While in typical experiments the lattice constant is shorter than the coherence length, we choose this slightly longer lattice constant to obtain a simple phase diagram containing large patches of topological phases. Moreover, in experiments the magnetic atoms are typically placed on the surface of a 3D superconductor, reducing the coupling between the Shiba states in comparison to our 2D model calculations. Therefore, the magnitudes of the effective couplings between the Shiba states in our model are probably quite realistic although we set the distance between the impurity atoms larger than in experiments. We have also included this paragraph in the manuscript.

    1.3 For our numerical experiments it is important, that the system size is larger than the localization length of the edge modes. In addition, we believe that \(24 \times \xi\) is a system size which does not pose a significant experimental challenge.

    1.4 The energy window we are considering for the local density of states is quite realistic. We are also considering quite large disorder strengths so that if experiments would achieve lower levels of disorder the performance could even be better.

  2. As you suggested, we have included clarifying sentences about the disorder potential and spin-orbit coupling at an earlier part of the manuscript. Overall, we would like to keep the separation of technical details in order to appeal to a broader audience who are not necessarily interested in the details of the models.

[1] J. Röntynen and T. Ojanen, Chern mosaic: Topology of chiral superconductivity on ferromagnetic adatom lattices, Phys. Rev. B 93, 094521 (2016).

---

## Round 2 · List of Changes

1. We have re-run our numerical experiments systematically and updated the figures and numbers accordingly. The results have slightly improved which we attribute to a \(\sim20\%\) increase in training dataset size.

  2. In the references, we provided a link to the source code that we used to generate the training data as well as to train and evaluate the neural network ensembles.

  3. In the references, we provided a link to the Shiba lattice model dataset on which we evaluated the performance of the neural network ensembles.

  4. In the introduction, we added a paragraph citing the works suggested by the editor along with some complementary references from the machine learning assisted quantum phase classification literature. We conclude this paragraph by explaining why our work is novel in this context.

  5. We added an explanation to the caption of Fig. 1, as to why the types of images we use as input to the neural networks can be expected in an experiment: "The LDOS is averaged over a symmetric energy window around zero energy \([-\overline{\Delta}_{\text{shiba}} / 6, \overline{\Delta}_{\text{shiba}} / 6]\), and therefore, the tunneling density of states is proportional to the quasiparticle density of states. Hence, STM measurements in the weak coupling limit will produce the type of pictures illustrated here."

  6. We now introduce the disorder potential scale \(V_0\) at an earlier point in the manuscript.

  7. We reformulated our claims in a more modest wording in abstract, introduction and discussion. In addition, we further highlighted the scope of our numerical experiments by adding "In the future it would be important to extend testing of the protocol beyond the single model and restricted parameter space used in this work." to the discussion section.

  8. We have expanded the discussion of our choice of model parameters and now explain that "The qualitative features of the phase diagram do not depend on the exact value of \(\lambda\) [1]." and argue that our system size is realistic, because we expect that "manufacturing systems of size \(24\xi\) does not pose a significant experimental challenge". With respect to the question why we set the superconducting coherence length equal to the impurity lattice spacing, we now clarify that "While in typical experiments the lattice constant is shorter than the coherence length, we choose this slightly longer lattice constant to obtain a simple phase diagram containing large patches of topological phases. Moreover, in experiments the magnetic atoms are typically placed on the surface of a 3D superconductor, reducing the coupling between the Shiba states in comparison to our 2D model calculations. Therefore, the magnitudes of the effective couplings between the Shiba states in our model are probably quite realistic although we set the distance between the impurity atoms larger than in experiments.".

  9. In our previous numerical experiments, we had used a slightly different threshold for the minimum bulk gap in training and test data which made the manuscript difficult to read. Therefore, we have unified this in our updated numerical experiments to \(3.5 / N_x \approx 0.15\).

  10. We have revised the methods section, added further technical details, and provided a reference to the source code that we used to train and evaluate the neural network ensembles.

  11. We fixed the spelling and formatting errors pointed out by the referee.

[1] J. Röntynen and T. Ojanen, Chern mosaic: Topology of chiral superconductivity on ferromagnetic adatom lattices, Phys. Rev. B 93, 094521 (2016).

---

## Editorial Decision

published